# Viral Co-Infection in Bats: A Systematic Review

**DOI:** 10.3390/v15091860

**Published:** 2023-08-31

**Authors:** Brent D. Jones, Eli J. Kaufman, Alison J. Peel

**Affiliations:** 1Centre for Planetary Health and Food Security, Griffith University, Nathan, QLD 4111, Australia; 2School of Environment and Science, Griffith University, Nathan, QLD 4111, Australia; 3Vassar College, Poughkeepsie, NY 12604, USA

**Keywords:** co-infection, coinfection, viral co-infection, viral dynamics, wildlife, chiroptera, review

## Abstract

Co-infection is an underappreciated phenomenon in contemporary disease ecology despite its ubiquity and importance in nature. Viruses, and other co-infecting agents, can interact in ways that shape host and agent communities, influence infection dynamics, and drive evolutionary selective pressures. Bats are host to many viruses of zoonotic potential and have drawn increasing attention in their role as wildlife reservoirs for human spillover. However, the role of co-infection in driving viral transmission dynamics within bats is unknown. Here, we systematically review peer-reviewed literature reporting viral co-infections in bats. We show that viral co-infection is common in bats but is often only reported as an incidental finding. Biases identified in our study database related to virus and host species were pre-existing in virus studies of bats generally. Studies largely speculated on the role co-infection plays in viral recombination and few investigated potential drivers or impacts of co-infection. Our results demonstrate that current knowledge of co-infection in bats is an ad hoc by-product of viral discovery efforts, and that future targeted co-infection studies will improve our understanding of the role it plays. Adding to the broader context of co-infection studies in other wildlife species, we anticipate our review will inform future co-infection study design and reporting in bats. Consideration of detection strategy, including potential viral targets, and appropriate analysis methodology will provide more robust results and facilitate further investigation of the role of viral co-infection in bat reservoirs.

## 1. Introduction

Co-infection with multiple infectious agents is common in vertebrate species; however, research often focuses on individual agents in isolation [1]. Although analytically straightforward, the single host-single agent paradigm fails to assess potential interactions that occur during co-infection. Facilitative and competitive dynamics between co-infecting agents have been reported [2,3,4], as has altered host susceptibility and disease severity due to co-infection [5,6,7]. This can directly influence important epidemiologic parameters, including the transmissibility of an agent, contact rate between hosts, and duration of infectiousness [8]. Neglecting to consider co-infection when investigating disease dynamics within a population risks missing important agent–agent relationships which may be acting as fundamental drivers of disease transmission within that population [8,9]. This has led to calls for increased consideration of co-infection within infectious disease research [1,10,11].

Bats are host to a diverse range of viruses, many of which are important zoonoses [12]. They are a unique mammalian host due to their ability to tolerate viral infections largely with no consequence, their ability to fly, and for many species, their highly gregarious nature [13]. Phylogenetic analysis has also suggested bats play an important evolutionary role for viruses, with many extant viruses from a diverse range of mammalian hosts having ancestral links to viruses originating in bats [14,15,16]. This makes an argument for viral research in bats particularly compelling. Not only are there implications for public health, but also for wildlife conservation and our broader understanding of viral evolution. 

Attempts to understand transmission dynamics within, and drivers of spillover from, bats have focused on seasonal, environmental, and host factors within single virus systems [17]. Advancements in the multiplexing capability of traditional detection assays, and the advent of next-generation sequencing (NGS) technologies has greatly enhanced our ability to screen for multiple viruses simultaneously [18]. Furthermore, NGS has reduced the necessity for prior knowledge of viral genomes, allowing the unbiased detection of the entire viral community within a sample [19]. This has resulted in a proliferation of studies exploring viral diversity within bats and attempts to characterise the bat-associated virome. When implemented without ecological context, or in isolation of downstream viral characterisation, viral discovery has been criticised for failing to realise tangible outputs in terms of spillover prevention and greater understanding beyond viral identity [12,20]. Recent virome studies across bat communities have provided insights into cross-species viral community composition [21,22]. Yet, co-infections within individual bats remains relatively unexplored. Review of the existing reports of multi-viral and unbiased viral screening studies is required to assess whether advancements in screening capacity have resulted in improved detection and understanding of co-infection as a process shaping viral community composition and transmission dynamics within chiropteran hosts.

This review aims to identify reports of viral co-infection in bats in the scientific literature; compare the occurrence and composition of co-infections across host and viral taxa involved; and identify broad trends in the detection, analysis, and discussion of co-infection within chiropteran hosts.

## 2. Materials and Methods

### 2.1. Literature Search Strategy

A search strategy involving six individual searches from eight databases (Appendix A) was performed on the 2 December 2021. Search strings incorporating synonyms from three key concepts, “bat”, “viral”, and “co-infection”, were tailored to each database format, and where the interface allowed, were further targeted using the following subject terms: bat, Chiroptera, virus, virology, infection, or infectious disease. All results were exported to EndNote 20.6 Build 17174 (Philadelphia, PA, USA) [23] reference management software and duplicates were removed before initial screening.

### 2.2. Screening and Eligibility Criteria

Initial screening by title and abstract was performed to remove records not describing empirical investigations of the viral infection of bats (e.g., reviews, in vitro studies, etc.). Following this, full texts of remaining English language records were retrieved to assess eligibility. To qualify for inclusion in the co-infection database, studies must have reported instances of natural co-infection of individual bats with viruses able to be classified as taxonomically distinct from each other. This may have been reported in written text, tables, graphics, or Appendix A published with the paper. Studies using any method of collection or processing that leads to pooled or combined results which cannot be definitively attributed to individuals were excluded. Additionally, studies relying on serology to detect infection were excluded as it identifies exposure rather than current infection, and so cannot be used to confirm co-infection. Publications that tested for multiple viruses but did not detect co-infection were included in an extended database, referred to as the full database, to allow for a comparison of sampling effort. Preprints were included. Studies were not assessed for risk of bias prior to data extraction. All screening was performed by a single author (BDJ). 

### 2.3. Data Extraction

Data extracted from studies reporting co-infection included the following: title, first author, year of publication, journal, co-infection terminology used and the section of the article in which it was mentioned (e.g., abstract, discussion, etc.), co-infection themes discussed, results of analysis of associations between co-infecting viruses and hosts, geographic location of sampling, host family sampled, number positive for single infection and co-infection, viruses tested for and detected, the detection methods used, and the full citation. Data extracted from studies assessing the presence of multiple viruses but not detecting co-infection included the following: title, year of publication, number of individuals sampled, host species sampled, viruses tested for and detected, and full citation.

### 2.4. Study-Level Comparison

Co-infected proportion (proportion of positive individuals that were co-infected) was determined for each study to summarise and allow comparison of co-infection within the database and across sampling location and detection assay used. Study results were then stratified by viral and host family to investigate whether rates of co-infection differed between viral and host families. The median and interquartile range (IQR) were used to compare groups. Missing data (e.g., host identity when co-infections were reported in text) meant that some studies were excluded from the study-level comparison. Assumptions regarding the results of each study and stratification process are included in Appendix A. 

### 2.5. Individual-Level Comparison of Pairwise Co-Detection across Viral and Host Families

Pairwise co-detection matrices were constructed to compare the screening and detection frequency of paired virus–virus and host–virus combinations. Each count within the matrices represents a single screening or detection event, and not a unique individual bat. As individuals may be tested for multiple viral families, they may contribute to multiple pairwise events. Counts in the host–virus detection matrix represent co-infection only and not all detections of viruses with hosts. 

### 2.6. Network Analysis of Co-Infection Themes of Discussion

Specific themes discussed in association with viral co-infection were identified during the data extraction process and recorded, including specific events preceding, involving, or related to co-infection. A network analysis was performed to visually identify trends in these discussion points. Themes were plotted as nodes, weighted so that the size of the node represented the number of publications discussing that theme, and edges weighted to reflect how frequently connecting nodes were discussed together. Themes were further grouped by similarity and role within the co-infection paradigm and nodes were coloured to reflect this grouping. 

All analyses and plots were performed using R Statistical Software 4.3.1 (Vienna, Austria) [24] in RStudio 2023.06.1 Build 524 (Boston, MA, USA) [25] using the tidygraph 1.2.3, ggraph 2.1.0, metbrewer 0.2.0, ComplexHeatmap 2.16.0, and the tidyverse 2.0.0 packages [26,27,28,29,30].

## 3. Results

### 3.1. Database Overview

The search identified 61 publications reporting viral co-infection in 662 individual bats [31,32,33,34,35,36,37,38,39,40,41,42,43,44,45,46,47,48,49,50,51,52,53,54,55,56,57,58,59,60,61,62,63,64,65,66,67,68,69,70,71,72,73,74,75,76,77,78,79,80,81,82,83,84,85,86,87,88,89,90,91], from 117 studies testing a total of 77,266 bats for multiple viruses (Figure 1) [16,31,32,33,34,35,36,37,38,39,40,41,42,43,44,45,46,47,48,49,50,51,52,53,54,55,56,57,58,59,60,61,62,63,64,65,66,67,68,69,70,71,72,73,74,75,76,77,78,79,80,81,82,83,84,85,86,87,88,89,90,91,92,93,94,95,96,97,98,99,100,101,102,103,104,105,106,107,108,109,110,111,112,113,114,115,116,117,118,119,120,121,122,123,124,125,126,127,128,129,130,131,132,133,134,135,136,137,138,139,140,141,142,143,144,145,146]. Publication of multi-viral studies in bats has increased since 2007 (mean of 7.67 per year, 95% CI: 4.45–10.88), as has the publication of studies detecting co-infection (mean of 3.93 per year, 95% CI: 2.14–5.73) (Figure 2). Screening coverage was high, including 85.7% (*n* = 30) of viral families known to infect bats, and 76.2% (*n* = 16) of recognised chiropteran families. However, co-infections were detected for only 54.3% (*n* = 19) of viral families and 52.4% (*n* = 11) of bat families. The number of viral families screened per study in the full database and in those studies reporting co-infection were skewed, as was the number of host families screened per study (Appendix A). Co-infection reports come from all continents where bats are found; however, a geographic bias was present in the database, with most studies that detected co-infection sampling bats in Asia (*n* = 32), followed by Europe (*n* = 12) and Africa (*n* = 11) (Appendix A). Most studies detected the presence of viruses using single assay strategies (*n* = 40), including either a form of PCR (e.g., conventional, reverse transcriptase, real-time, etc.) (*n* = 36) or NGS (*n* = 4). Multiple assay strategies were used in the remaining studies, including combining PCR and NGS (*n* = 10); PCR and viral isolation (*n* = 8); or PCR, NGS, and viral isolation (*n* = 3). PCR remained the most frequently used assay over time and viral isolation remained consistently low, whereas NGS saw an increase in 2020 and 2021 (Appendix A).

### 3.2. Sampling Effort vs. Detection

Sampling effort was biased towards RNA viruses at a study (89.7%; *n* = 105) and individual (91.9%; *n* = 70,987) level compared with DNA viruses (33.3%, *n* = 39 studies; 29.1%, *n* = 22,465 individuals). Detection of co-infection mirrored this at a study level, with RNA viruses being reported more often (82%; *n* = 50) than DNA viruses (27.9%; *n* = 17), though this was less apparent at an individual level (RNA 55.3%; *n* = 366, and DNA 48%; *n* = 318). 

Unsurprisingly, viral families with an RNA genome dominated database reports as detection followed sampling effort for the most part (Figure 3a and Appendix A). *Coronaviridae* were tested for most at a study (47.9%; *n* = 56) and individual (48.9%; *n* = 37,823) level and reported co-infection in the most studies (45.9%; *n* = 28) and the second most individuals (29.7%; *n* = 197). *Herpesviridae* reported the most co-infected individuals (34.3%; *n* = 227), despite less than 10% of individuals in the full database being screened for the viral family (*n* = 6058). *Picornaviridae* were also overrepresented in co-infections at an individual level (10.7%; *n* = 71 co-infections compared to 7.1%; *n* = 5508 tested). Both anomalies can be explained by a single paper reporting high numbers of co-infection for each viral family [42,86]. *Rhabdoviridae* are underrepresented in co-infected individuals (0.7%; *n* = 5), despite a high sampling effort (23.1%; *n* = 27 studies, and 22.6%; *n* = 17,450 individuals), and this is explained by low study prevalence within the database.

The comparison of sampling effort between chiropteran suborders was relatively even at both the study (Yangochiroptera, 78.6%, *n* = 92; Yinpterochiroptera, 61.5%, *n* = 72) and individual (Yangochiroptera, 37.4%, *n* = 28,869; Yinpterochiroptera, 49.3%, *n* = 38,091) levels (Figure 3b and Appendix A). However, this contrasts with the species diversity within each suborder, as Yangochiroptera accounts for most chiropteran species (70.9%; *n* = 1036) compared with Yinpterochiroptera (29.1%; *n* = 426) [147]. Detection followed sampling effort, with Yangochiroptera reporting most at a study level (59%; *n* = 36) compared to Yinpterochiroptera (42.6%; *n* = 26), but not at an individual level (37.2%; *n* = 249 vs. 52.9%; *n* = 350). At a host family level, detection was again consistent with sampling effort (Figure 3b). Vespertilionidae (61.5%, *n* = 72 studies screening; 26.2%, *n* = 16 detecting; 21.6%, *n* = 16,666 individuals screened; 23%, *n* = 152 individuals co-infected) and Pteropodidae (43.6%, *n* = 51 studies screened; 27.9%, *n* = 17 detecting; 19.6%, *n* = 15,152 individuals screened; 38.5%, *n* = 255 individuals co-infected) feature the most in terms of sampling and detection at both the study and individual level.

### 3.3. Co-Infected Proportion

The median co-infected proportion across all studies detecting co-infection was 0.11 (IQR 0.04–0.19; *n* = 59). This varied by continent; however, IQRs largely overlapped (Figure 4a). Asia reported the narrowest IQR and second lowest median (0.10; IQR 0.04–0.16; *n* = 32), while Europe reported the highest median, though had the second widest IQR (0.24; IQR 0.05–0.38; *n* = 12). Australia (0.13; IQR 0.11–0.56; *n* = 3), South America (0.14; IQR 0.11–0.27; *n* = 4), and Africa (0.06; IQR 0.03–0.18; *n* = 11) were comparable. North America was excluded due to too few studies (*n* = 2).

When comparing across detection assay strategy, studies using NGS alone reported the highest median (0.40; IQR 0.19–0.57), followed by studies using PCR, viral isolation, and NGS (0.24; IQR 0.17–0.29). Studies using PCR alone (0.09; IQR 0.03–0.15), PCR and NGS (0.10; IQR 0.04–0.19), and PCR and viral isolation (0.09; IQR 0.04–0.19) had almost identical medians (Figure 4b). More broadly, there appeared to be no difference comparing studies using multiple assays (0.11; IQR 0.05–0.24) to those relying on single assay types alone (0.10; IQR 0.04–0.18).

Comparison of viral families was restricted to nine families and appeared more dependent on number of studies than viral family identity. Those reported in more than five studies exhibited lower medians and tighter IQRs. *Herpesviridae* reported the lowest (0.06; IQR 0.03–0.24), followed by *Coronaviridae* (0.08; IQR 0.03–0.16), *Astroviridae* (0.14; IQR 0.20–0.38), *Paramyxoviridae* (0.27; IQR 0.20–0.38), and *Adenoviridae* (0.29; IQR 0.20–0.48), while *Polyomaviridae* (0.25; IQR 0.09–1), *Picornaviridae* (0.78; IQR 0.45–1), *Reoviridae* (1; IQR 0.4–1), and *Rhabdoviridae* (1; IQR 0.5–1) had much higher medians and/or IQRs (Figure 4c).

Comparison of co-infections among bat hosts in seven host families with sufficient studies to allow comparison indicated that rates of co-infection proportions were highest in Vespertilionidae (0.27; IQR 0.06–0.38), followed by Hipposideridae (0.25; IQR 0.14–0.25), Molossidae (0.18; IQR 0.16–0.19), Miniopteridae (0.13; IQR 0.07–0.35), Phyllostomidae (0.10; IQR 0.04–0.21), Pteropodidae (0.09; IQR 0.04–0.27), and Rhinolophidae (0.07; IQR 0.05–0.10) (Figure 4d).

### 3.4. Pairwise Detections

The mean number of viral families an individual viral family was tested against in the database (across all studies combined) was 16.55 (95% CI: 13.96–19.13) and ranged from 29 different viral families being tested for co-infection with *Adenoviridae*, to 5 with *Phenuiviridae* (Appendix A). The mean number of viral families an individual viral family was detected co-infected with was 5.05 (95% CI: 3.60–6.50) and ranged from 11 for *Coronaviridae* to 1 for *Retroviridae*, *Filoviridae*, *Phenuiviridae*, and *Flaviviridae* (Appendix A). A total of 276 unique pairwise viral–viral combinations amounting to 579,227 testing events were assessed in the database. This can be further divided into 250 unique interfamilial combinations (*n* = 473,502 events), and 26 intrafamilial combinations (*n* = 105,725 events) (Figure 5a). Positive co-infection events were reported for 57 unique combinations (*n* = 747 events), with 13 being intrafamilial (*n* = 515 events), and 44 being interfamilial (*n* = 232 events) (Figure 5b). Focusing on those viral families involved in greater than 10% of co-infection reports, four report intrafamilial detection in the greatest proportion: *Herpesviridae* (63%; *n* = 217), *Coronaviridae* (49.55%; *n* = 111), *Picornaviridae* (93.24%; *n* = 69), and *Paramyxoviridae* (39.80%; *n* = 41). Pairwise detection with *Coronaviridae* account for the most co-infections for *Adenoviridae* (35.29%; *n* = 30) and *Astroviridae* (49.45%; *n* = 45) (Figure 5b).

The mean number of viral families a host family was tested against in the database (across all studies combined) was 16.25 (95% CI: 10.68–21.32) and ranged from 32 for Vespertilionidae to 1 for both Craseonycteridae and Noctilionidae (Appendix A). The mean number of viral families an individual host family was detected co-infected with was 5.18 (95% CI: 2.63–7.73) and ranged from 15 for Vespertilionidae to 1 for Mormoopidae (Appendix A). A total of 260 unique pairwise host–viral combinations amounting to 169,500 testing events were assessed in the database (Figure 6a), of which 720 events from 58 combinations were positive for co-infection (Figure 6b). The host family was unknown for 10,306 individuals which were tested for 19 possible combinations (*n* = 19,610 events), of which 5 combinations (*n* = 94 events) were positive for co-infection. 

Focusing on those host families involved in greater than 5% of co-infection reports, *Coronaviridae* made up the most co-infections for Rhinolophidae (91.52%; *n* = 54) and Miniopteridae (81.63%; *n* = 40). Pteropodidae were reported most co-infected with *Herpesviridae* (75%; *n* = 198), and Vespertilionidae were coinfected most with *Picornaviridae* (31.94%; *n* = 69), followed by *Coronaviridae* (18.51%; *n* = 40) and *Adenoviridae* (15.74%; *n* = 34). Finally, *Paramyxoviridae* accounted for the most co-infections with Phyllostomidae hosts (65.85%; *n* = 27), followed by *Herpesviridae* (24.39%; *n* = 10) (Figure 6b).

### 3.5. Measures of Association

Five papers sought to determine potential associations between co-infecting viruses using statistical measures, of which four were successful (Appendix A). Positive associations were identified between coronaviruses and astroviruses, paramyxoviruses and astroviruses; and adenoviruses and rhabdoviruses. There are, however, inconsistencies between reports. Seltmann et al. (2007) reported an association between co-infecting coronaviruses and astroviruses, while Chu et al. (2008), Anthony et al. (2013), and Hoarau et al. (2021) were unable to demonstrate such a relationship [33,42,62,82]. Similarly, Hoarau et al. (2021) identified a positive relationship between paramyxoviruses and astroviruses, while Anthony et al. (2013) did not [42,82]. Anthony et al. (2013) found both positive and negative associations between differing strains of herpesviruses [42]. It was the only paper to classify below the taxonomic level of family, and in doing so, attempt to measure intrafamilial associations. Methods varied and included Chi square tests (*n* = 2) [33,82], generalised linear models (*n* = 2) [53,62], and a Bayesian approach to measuring cooccurrence using the *C* score statistic (*n* = 1) [42]. A single study investigated potential associations between host family and the rate of co-infection, and another assessed the impact seasonal and geographic differences may have on the rate of co-infection, but both failed to find evidence of an association [82,88].

### 3.6. Terminology and Themes

Almost a quarter of publications detecting co-infection make no specific reference to that co-infection within the text (*n* = 13). Of the remaining papers that do use specific terminology to report co-infection, few mention it in the abstract (*n* = 15; 33%), or title and keywords (*n* = 4; 9%). Twelve unique expressions were used to describe co-infection, with co-infection (53%) and coinfection (29%) being the most frequently used (Appendix A). Twenty-five papers (43%) discuss co-infection in the broader context of viral ecology and epidemiology. Eighteen themes of discussion were identified and grouped within three broad categories: those conditions required or likely to promote co-infection (termed Drivers), the consequences of viral co-infection (termed Outcomes), and the mechanisms occurring between co-infecting viruses (termed Mechanisms). Outcomes of co-infection dominated the discussion, with recombination (*n* = 15), host spillover (*n* = 9), and emergence (*n* = 6) the most commonly occurring and most frequently discussed together within the network graph (Figure 7). Host population density and host restriction (*n* = 3) were the most frequently mentioned input themes. Interaction (*n* = 4) was the mechanism theme most discussed and was used in the context of impacting recombination, shedding, transmission dynamics, and host fitness (Figure 7).

### 3.7. Co-Infection of Viral Families with High Emergence Potential

An area of significant focus in bat virology is the spillover or emergence potential of viruses, particularly those in the *Coronaviridae*, *Paramyxoviridae*, *Filoviridae*, and *Reoviridae* families [148]. This interest is reflected in the database with coronaviruses and paramyxoviruses reported in the largest number of studies, though reoviruses and filoviruses featured in fewer studies (Figure 3a). Co-infection was detected in all families, with intrafamilial co-infection most frequently reported (Figure 5b); however, the absolute number of co-infected bats reported varied substantially between the families (*Coronaviridae*, *n* = 197; *Paramyxoviridae*, *n* = 66; *Reoviridae*, *n* = 9; and *Filoviridae*, *n* = 1). Acknowledging large gaps in the screening effort across host and viral families (Figure 6a), most detections of *Coronaviridae* co-infections were in *Rhinolophidae*, *Hipposideridae*, and *Miniopteridae*, and most *Paramyxoviridae* co-infections were in *Phylostomidae* and *Pteropodidae* (Appendix A).

## 4. Discussion

As with any complex ecological community, many potential interactions exist within communities of infectious agents and their hosts [9,149]. Infectious agents occupy different host species and tissue niches within a host and exhibit a variety of excretion pathways. This results in a variety of sample types to enable detection with. These infections are rarely static in individuals or populations across space and time, making it challenging to accurately describe infections in wild animal populations at any given time or infer characteristics of infection dynamics more broadly. These challenges are amplified for the detection of co-infections and the interactions among them, yet there is a growing recognition of the importance of viral co-infections in driving viral evolution and ecology and host responses to infection [9,149,150]. Considering this, and the increase in focus in recent years on the potential public health risk of the emergence of zoonotic viruses from wild bat populations, we sought to critically analyse the existing literature on viral co-infections in bats. Through this analysis, we aimed to identify key knowledge gaps that warrant further research.

We detected an increase in viral co-infection survey effort in the past 15 years, supported by an increase in the number of studies and absolute number of individual bats screened. This increase is likely a by-product of the expanding viral discovery effort in bats generally [151]. Growing interest in the role bats play as wildlife viral reservoirs and advancements in molecular-based detection technologies are cited as driving viral discovery efforts, and likely have the flow-on effect of increasing multi-viral studies and reports of co-infection [12,151]. We conclude that co-infections were rarely the primary research goal within the studies in our datasets. This conclusion is supported by three key results: the biases evident in screening effort, the restricted use of specific terminology, and the discussion trends reported. 

Firstly, we found that screening effort in multi-viral surveys of wild bats was biased across host and viral families, and detection methods in a manner similarly reported in publicly available sequence databases and bat viral publications more broadly [12,152,153]. For example, effort was skewed towards screening RNA viruses of zoonotic potential (e.g., *Coronaviridae*, *Paramyxoviridae*, and *Rhabdoviridae*). This mirrors the bat-borne viral discovery effort generally, as RNA viruses account for 85% of the approximate 16,600 bat-associated viral sequences on GenBank, with coronaviruses, rhabdoviruses, and paramyxoviruses making up 30%, 24%, and 10%, respectively [153]. Screening effort across host families was biased towards the suborder Yinpterochiroptera, driven by the high sampling of bats from the Pteropodidae, Rhinolophidae, and Hipposideridae families, despite the suborder representing just under 30% of all bat species diversity [147]. Such screening bias has been reported previously [151]. Furthermore, studies reporting co-infection did not reflect the geographic distribution of bat taxonomic diversity. A parsimonious explanation for this is the preference for studies on viruses of public health significance, such as coronaviruses and paramyxoviruses, in bats sampled in Asia, following the emergence of SARS-CoV-1 and the Nipah virus [154,155]. 

Secondly, the inconsistent and restricted use of terminology for co-infection in publications reporting co-infection is an indication of the lower priority co-infection has on the agenda of bat researchers. This was notably demonstrated by our finding that 62% of papers detecting co-infection do not mention it in the title, abstract, or keywords, and 22% do not mention it at all in the main text. Finally, the discussion on co-infections was limited and focused on outcomes closely related to coronavirus ecology, for example, recombination, spillover, and emergence, while potential drivers and their relative contribution to co-infection remain speculative [31,37,49,50,51]. In contrast to broader reviews of co-infection dynamics in other systems [10,149,156], we found that mechanisms through which potential interaction may occur were discussed in general terms (e.g., association or competition), rather than specific pathways (e.g., immune system pathways, resource competition, etc.) [42,59,62,68,82]. This suggests that the field is simply in its infancy in bat research. However, the innocuous nature of viral infection in bats compared to other vertebrate hosts may also impact the discussion of viral co-infection [152,157]. This may further explain why viral recombination is the focus of co-infection discussions for bats, despite a plethora of examples of co-infection impacting host fitness and disease transmission dynamics in other host systems [6,158]. 

This review is the first to attempt to summarise the occurrence of viral co-infection in bats and explore how it differs across study level factors. However, as with studies of co-infection in humans and other animals, the ad hoc nature of the studies presented here limits meaningful meta-analyses by viral and host family, or geographic location [159]. Instead, this review provides opportunities for descriptive assessments of the variety of co-infections reported and the hypothesised interactions and outcomes for both host and agents. The median study co-infection proportion reported (0.11 in studies detecting co-infection) likely underestimates the true prevalence of co-infection. Median co-infected proportion is sensitive to outliers when the number of studies is small and when the number of positive individuals in a study is low. An example would be a co-infection between a virus typically detected at low prevalence (e.g., rhabdoviruses; [160]) and another typically detected at high prevalence (e.g., herpesviruses; [52,73,94]). A single co-infected individual screened for these two viral families would suggest that co-infection is common in rhabdovirus-infected individuals but rare in herpesvirus-infected individuals. Most studies only tested for a limited number of viral families, potentially missing co-infections with and among unstudied families. Our review findings provide little evidence for viral- or host-family-level drivers for co-infection in bats generally, though they were limited by most studies focusing on the same few viral families.

Technological advancement offers hope for gaining future insights into co-infections across a broader range of viral families. Indeed, trends in detection assay usage in multi-viral studies were consistent with what has been previously reported for viral discovery studies in bats [151]. Viral isolation efforts are declining, presumably due to labour intensity, logistic difficulty, and a limited availability of appropriate cell lines [161], while the increasing affordability and detection capacity of NGS has resulted in its increased use in metagenomic studies. Much of the discourse around detecting co-infections advocates the use of NGS as the ‘gold standard’ [149,162,163]. Our results tend to support this; however, the number of studies was limited compared to other PCR approaches. Recent studies comparing NGS and PCR found discrepancies in assay results, leading the authors to recommend a combined approach using multiple modalities [83]. Sensitivity and specificity for detecting a given viral family is also affected by the sample type associated with various viral excretion routes [164]. Studies using antemortem sample collection require a virus to be actively shedding, meaning latent infections or those in their eclipse phase will not be detected and may result in some co-infected individuals being false negatives. Competitive interactions can also decrease the amount of virus shed from a co-infected individual [3], potentially affecting detection in low sensitivity assays. The development of a framework to systematically design co-infection studies in the context of available detection methods and known sampling and testing limitations is required to maximise the utility of future co-infection studies in bats.

An ultimate objective for future co-infection studies in bats will be determining associations between co-infecting viruses, which, in turn, is a key step in identifying potential interactions and understanding how co-infection shapes viral transmission dynamics within the bat reservoir. Competition may arise between viruses via processes such as interferon-mediated immunity or resource competition [165,166]. Alternatively, facilitation can occur through immune suppression, or upregulated viral replication [167,168]. In general, however, we found scant evidence of pairwise associations at the level of viral families, with the viral composition of co-infections in our database being instead consistent with testing effort. The taxonomic designation of ‘family’ was a pragmatic approach to a diverse dataset here, but may be an inappropriate level at which to compare co-infecting viruses. Asymmetric associations between individual members of separate viral families could result in inconsistent estimates of association when measured at the family level. Evidence of this is reported in Anthony et al., 2013, where closely related strains of herpesvirus demonstrated both positive and negative associations [42]. The relationship between each viral pair is also likely to be conditional on seasonal, host, or geographic factors, which differ between each study, making comparison difficult, if not impossible. Further to this, differences in the broader viral community in each study may introduce further conditions if the presence of a third, unmeasured virus impacts the strength or direction of the original association. This has prompted the adoption of community ecology approaches to disentangling such complex relationships for other host–agent systems [169,170,171,172]. Utilising a lower taxonomic classification and methods appropriate for the complexity of co-infection data is recommended for the future investigation of associations between co-infecting agents.

The role co-infection may have in influencing the zoonotic potential of viruses circulating in bat populations is likely to remain a predominate area of focus as the field matures. A study published after our literature search was conducted utilized NGS to identify viruses the authors suggest have zoonotic emergence potential due to their close genetic relationship with existing human pathogens [21]. Consistent with the themes identified in the reviewed literature here, the co-infection and cross-species transmission identified was proposed as presenting opportunities for recombination and spillover from the bat reservoir [21]. While specific studies to determine this are required, the study illustrates how meta-transcriptomic approaches using NGS could advance our understanding of the role of co-infection in viral ecology and evolution, and the potential implications this may have for public health risk due to emerging viruses from the bat wildlife reservoir.

## 5. Conclusions

Our review provides an overview of the reporting of viral co-infection in bats globally using descriptive statistics and offers recommendations for future studies. Viral co-infection is commonly detected in viral studies of bats, albeit at a relatively low prevalence. We report some differences in co-infection rates between viral families; however, comparison is limited by testing bias. Discussion around co-infection in bats focuses on recombination and spillover, though limited effort has been directed at exploring the drivers of co-infection and potential associations between co-infecting viruses. We advocate for the use of the term co-infection in preference to other terms due to its dominance in the existing literature, and for its inclusion as a keyword or in the abstract of studies where the study design allowed for the detection of co-infections, whether they were detected or not. We encourage researchers to consider flexible approaches to analysis that allow for the consideration of conditional dependencies of viral associations and using the lowest taxonomic classification feasible. Finally, we identified current deficits in viral co-infection research, that if addressed, can improve our understanding of the role of co-infection in viral dynamics in chiropteran populations.

## Figures and Tables

**Figure 1 viruses-15-01860-f001:**
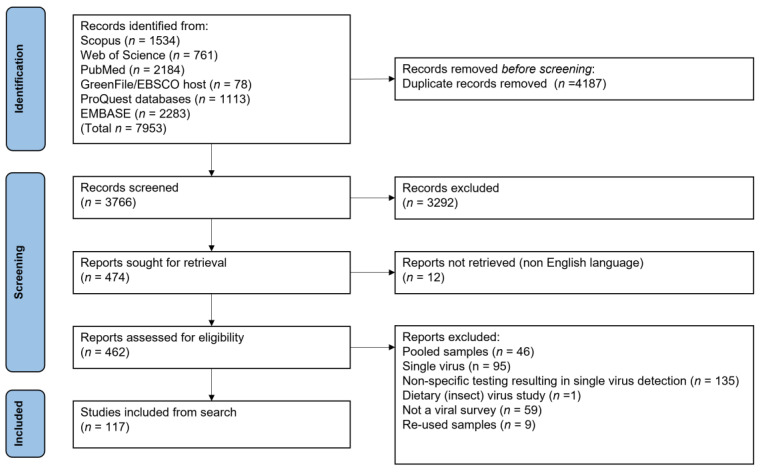
PRISMA flowchart depicting each step of the literature search and screening process. Full detail of all 117 studies included from the search provided in Appendix A.

**Figure 2 viruses-15-01860-f002:**
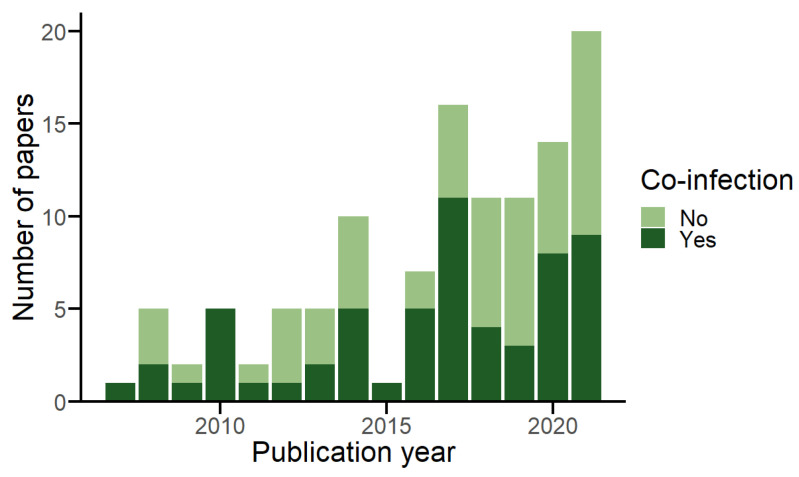
Publication of multi-viral studies in bats over time, showing the number of papers across the full database, grouped by whether co-infection was detected or not.

**Figure 3 viruses-15-01860-f003:**
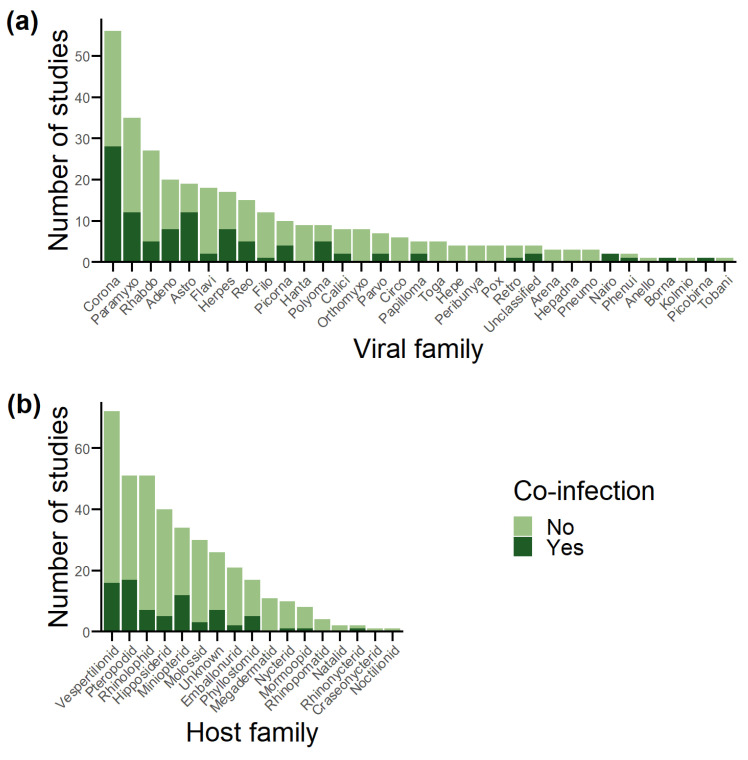
Number of publications involving each viral and bat host family in the database, grouped by whether co-infection was detected or not. (**a**) Viral families. (**b**) Host families.

**Figure 4 viruses-15-01860-f004:**
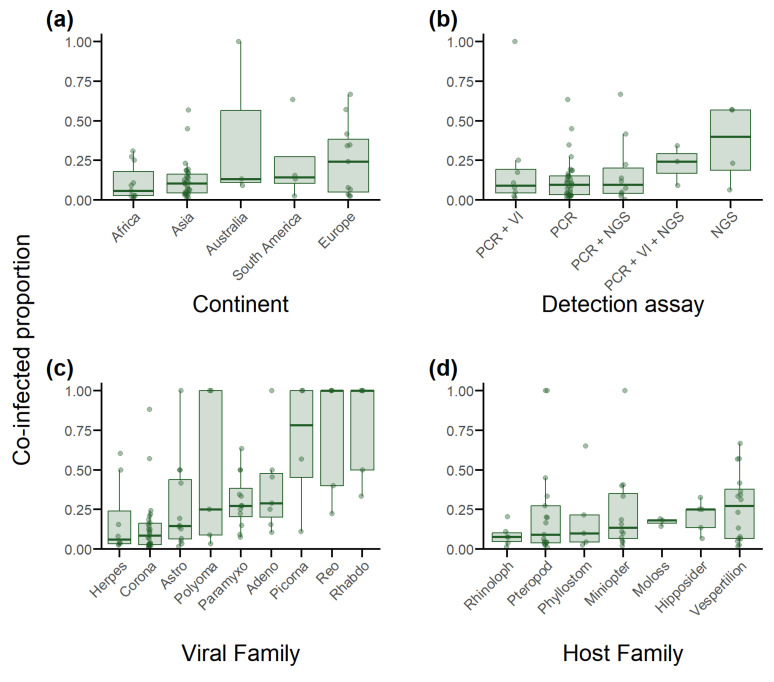
Boxplots depicting the median co-infected proportion and interquartile range described by the 25th and 75th percentiles of all studies in the database that detected co-infection, stratified by the continent the bats were sampled from, the viral detection strategy used (PCR, viral isolation (VI) or next-generation sequencing (NGS)), and the individual viral and host family. Co-infected proportion represents the proportion of positive individuals from each stratum that were co-infected. (**a**) Sampling location (continent). (**b**) Detection assays used. (**c**) Viral family. (**d**) Host family.

**Figure 5 viruses-15-01860-f005:**
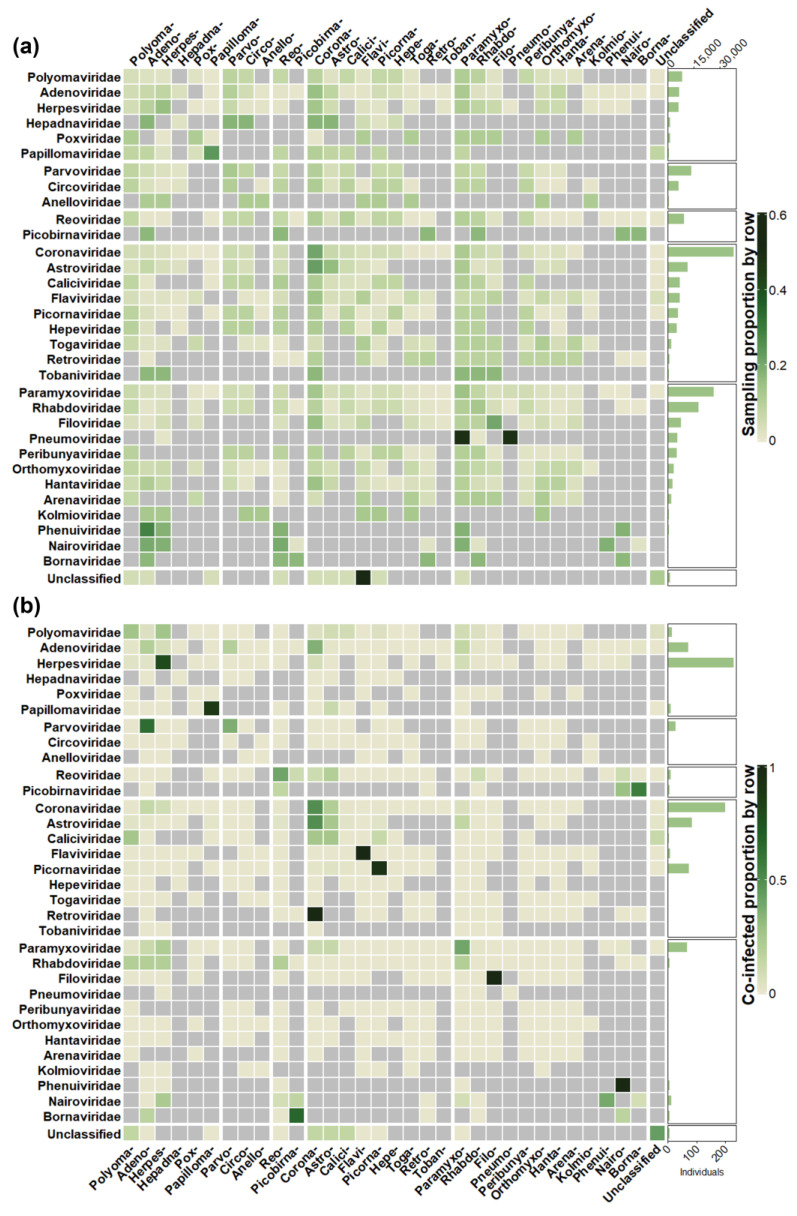
Heat map displaying the screening effort and detection of each pairwise viral family combination recorded in the database. The bar chart on the right displays the number of individual bats tested or infected with the viral family for each row. The heat map then displays the proportion of those individuals tested or infected with the viral family on each column. Values used to generate the plot are from Appendix A. (**a**) Screening effort. (**b**) Co-infections detected. Grey cells are combinations not tested for. Columns and rows are grouped by virus genome structure. The same viral families, in full and abbreviated form, are on the x and y axis, in identical order.

**Figure 6 viruses-15-01860-f006:**
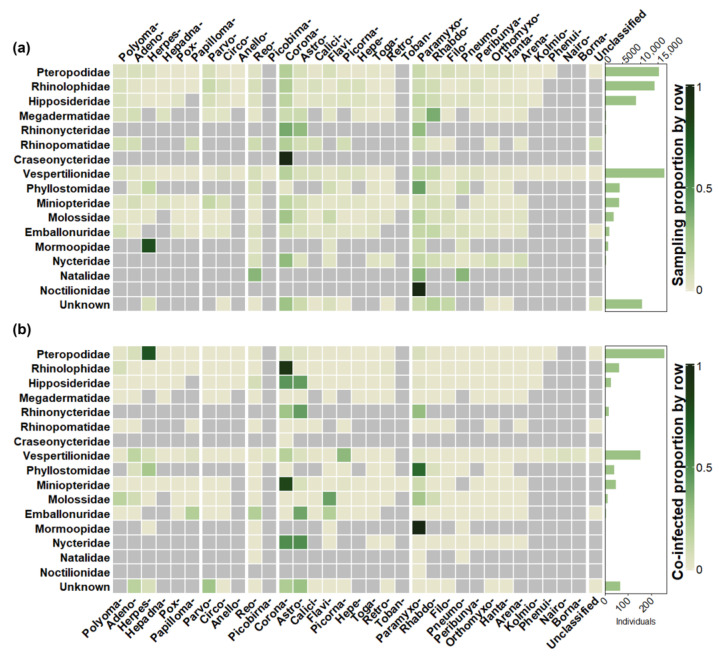
Heat map displaying the screening effort and detection of each pairwise host and viral family combination recorded in the database. The bar chart on the right displays the number of individual bats tested or infected from each host family for each row. The heat map then displays the proportion of those individuals tested or co-infected with the viral family on each column. Values used to generate the plot are from Appendix A. (**a**) Screening effort. (**b**) Co-infections detected. Grey cells are combinations not tested for. Columns and rows are grouped by virus genome structure. The same viral families, in abbreviated form, are present in Figure 5.

**Figure 7 viruses-15-01860-f007:**
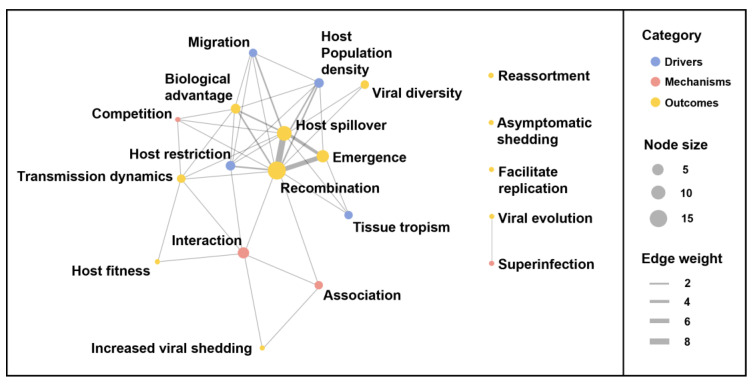
Network plot of themes discussed in the database. Each node is labelled with the description of a theme discussed in the publications in the database. The size of the node illustrates the number of publications discussing the corresponding theme, and the colour of the node corresponds to the broader classification the theme belongs to: drivers, mechanisms, outcomes. Themes that are discussed together in the same publication are connected by a line (edge). The thickness of the edge corresponds to the number of publications discussing the connected themes together.

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
