# Peer review of "Viral Co-Infection in Bats: A Systematic Review"

_viruses, 2023, doi:10.3390/v15091860_

Round 1

Reviewer 1 Report

The manuscript under consideration, Review: Viral coinfection in bats: a systematic review, is overall written well, but some areas need further improvement, to fetch high citations in the coming days.  

The comments and suggestions are as under: 

The thematic graphical abstract is missing. 

A comprehensive flow chart is recommended. 

The manuscript must include a heading with the title "Shared Viral Species among the Bats."

belonging to the Coronaviridae, Reoviridae, Picornaviridae, Parvoviridae and Polyomaviridae.

Another very important area that must be part of the manuscript, is " viruses of potentially high emergence risk" With the help of Phylogenetic analysis we can come up with viral species that were closely related to known human or livestock pathogens, which can be denoted.

“Viruses of concern” mainly viral families—the Coronaviridae 

and the Reoviridae). (, Bat SARS-like coronavirus CX1 was

detected in Rh. pusillus and Rh. marshalli, while Bat SARS-like coronavirus

LS1 was detected in Rh. macrotis and Rh. thomasi.)

The prevalence

of these viruses of concern is relatively high, especially Bat,

orthoreovirus BS1 and Bat rotavirus A type CX1.

The three coronaviruses were closely related to known zoonotic viruses that infect humans or swine, based on the protein "RdRp protein").

" Cross-species transmission of viruses among the bats" correlation of virome composition, based on host phylogeny and composition variation, (Bray–Curtis’s distance, Poisson regression)

For details regarding comments and include the following recent reference.

Wang, Jing, et al. "Individual bat virome analysis reveals co-infection and spillover among bats and virus zoonotic potential." Nature Communications 14.1 (2023): 4079.

Shi, M. et al. Total infection characterization of respiratory infections

in pre-COVID-19 Wuhan, China. PLoS Pathog. 18, e1010259

(2022).

Temmam, S. et al. Bat coronaviruses related to SARS-CoV-2 and

infectious for human cells. Nature 604, 330–336 (2022).

Minor English language and orcid numbers are missing 

Reviewer 2 Report

The manuscript by Jones et al. raises a very important question about the insufficient knowledge of the phenomenon of co-infection by various viral pathogens of the same host. The review gives an overview of the current knowledge of co-infection in bats and thus this work merits publication. Overall, this is quite a well-written and well-designed review article.

Currently, researchers are beginning to pay attention to co-infection and its impact on viral dynamics and the evolution of viruses. For example, in the publication "SARS-CoV-2 infection enhances by human adenovirus or influenza A virus in animal models" shows that both influenza and HAdV5 cause more severe diseases when co-infected with SARS-CoV-2. In the article in Lancet (Swets et al., 2022) it is written: “SARS-CoV-2 co-infections with influenza viruses and adenoviruses were each significantly associated with increased odds of death”. Undoubtedly, the role of co-infection in the development of viral epidemics requires further study.

Round 2

Reviewer 1 Report

The following comments needed to address: 

Unable to locate the graphical abstract: (https://www.google.com/search?q=graphical+abstract+Viral+infection+in+bats&tbm=isch&ved=2ahUKEwivgtTbsv6AAxULrycCHQeRDxMQ2-cCegQIABAA&oq=graphical+abstract+Viral+infection+in+bats&gs_lcp=CgNpbWcQA1CKLFiKLGDuMmgAcAB4AIABwgGIAYMDkgEDMC4ymAEAoAEBqgELZ3dzLXdpei1pbWfAAQE&sclient=img&ei=YRLsZO_NEYvensEPh6K-mAE&bih=712&biw=1455)

A comprehensive flow chart is recommended.

A flow chart of the systematic review as per the PRISMA guidelines for literature reviews is already included (Fig. 1). 

In the last box, explain the detail of 117 studies ? 

Minor editing of English language required
